# The Expanding Role of HLA-E in Host Defense: A Target for Broadly Applicable Vaccines and Immunotherapies

**DOI:** 10.3390/cells14241983

**Published:** 2025-12-14

**Authors:** Mahsa Rafieiyan, Marco Pio La Manna, Francesco Dieli, Nadia Caccamo, Giusto Davide Badami

**Affiliations:** 1Central Laboratory of Advanced Diagnosis and Biomedical Research (CLADIBIOR), University of Palermo, 90133 Palermo, Italy; mahsa.rafieiyan@unipa.it (M.R.); marcopio.lamanna@unipa.it (M.P.L.M.); nadia.caccamo@unipa.it (N.C.); giustodavide.badami@unipa.it (G.D.B.); 2Department of Health Promotion, Mother and Child Care, Internal Medicine and Medical Specialties (ProMISE), University of Palermo, 90133 Palermo, Italy; 3Department of Biomedicine, Neuroscience and Advanced Diagnosis (BiND), University of Palermo, 90133 Palermo, Italy

**Keywords:** T cell responses, HLA-E, immunotherapy

## Abstract

Human leukocyte antigen (HLA)-E, a non-classical class I molecule with limited polymorphism, bridges innate and adaptive immunity. Traditionally, the role of HLA-E had been associated with regulating natural killer (NK) cell activity via CD94/NKG2 receptors, by presenting self-peptides derived from the leader sequence of HLA-I. Recent findings reveal its ability to present pathogen-derived peptides to CD8^+^ T cells, eliciting unconventional cytotoxic responses. This review examines the expanding role of HLA-E-restricted T cells in viral and bacterial infections and their capacity to recognize diverse microbial peptides and enhance immune response when classical HLA pathways are impaired. We also highlight key advances in immunotherapy and vaccine development, including CMV-vectored platforms, donor-unrestricted TCR-based strategies, and peptide prediction algorithms. The minimal polymorphism of HLA-E, its resistance to viral immune evasion, and its ability to present conserved pathogen peptides position it as a promising target for universal vaccines and next-generation immunotherapies. Understanding these unconventional roles may pave the way for broadly applicable immunotherapies and vaccines against infectious diseases.

## 1. Introduction

T cell receptors (TCRs) expressed by CD8^+^ T lymphocytes recognize specific endogenous peptides presented by major histocompatibility complex (MHC) class I molecules on the cell surface. The specificity of the interaction between the MHC class I and TCRs has been exploited in immunotherapeutic applications, particularly in the development of innovative vaccines. However, TCR-based biologics targeting infectious antigens face limitations in clinical applications due to the diversity of MHC class Ia molecules. Classical MHC class Ia molecules (in humans, Human Leukocyte Antigen la) can bind and present a wide array of peptides to T cells and activate them, yet they exhibit substantial genetic variation [1,2,3]. In fact, HLA molecules display remarkable polymorphism, with 28,409 classical HLA class I (HLA class Ia) alleles identified as of 15 January 2025 [4,5,6,7,8].

In contrast, non-classical HLA class I molecules such as HLA-E exhibit limited polymorphism and are relatively conserved [9], offering a more universal therapeutic approach. Moreover, HLA-E can modulate both innate and adaptive immunity through interaction with inhibitory and activating receptors. This review explains the interplay between canonical and pathogen-derived epitopes presented by HLA-E and their recognition by innate and adaptive immune receptors, and how it can lead to enhanced or evaded immune responses. 

HLA-E and HLA-G molecules regulate the immune response. In contrast to the HLA-E, HLA-G mainly provides inhibitory signals to natural killer (NK) cells, facilitating immune escape in cytomegalovirus (CMV), Hepatitis B Virus (HBV), and Hepatitis C Virus (HCV) infections. Regarding adaptive immunity, HLA-E presents pathogen-derived peptides and elicits specific CD8^+^ T cell responses, whereas HLA-G expression during viral infections correlates with immunosuppression and viral persistence. HLA-G primarily mediates immune tolerance and, by presenting viral peptides, could evade immune detection and impair pregnancy [10,11].

Recent studies highlight that HLA-E’s minimal polymorphism, ability to present conserved pathogen peptides, resistance to viral immune evasion, and dual role in NK and CD8^+^ T cell activation make it a highly effective platform for universal vaccine development. It also highlights the current limited understanding of T cell recognition of HLA-E–presented peptides across diverse infections and explores how advancing this knowledge could inform the design of future HLA-E–based vaccines and immunotherapies targeting microbial pathogens. 

We conducted a systematic search of Scopus, PubMed, and Web of Science for peer-reviewed English-language articles published between 1998 and 2025. Search terms combined “HLA-E,” “MHC,” “pathogen,” “NK cell,” “CD8^+^ T cell,” and “vaccine.” Recent high-impact studies were prioritized, alongside seminal works for historical and mechanistic context. Non-English and non-peer-reviewed articles were excluded.

## 2. Characteristics of HLA-E

The HLA-E molecule is encoded by only two functional alleles, HLA-E*01:03 and HLA-E*01:01. These alleles are translated into proteins that differ by a single amino acid at position 107: HLA-E*01:03 has glycine (Gly), whereas HLA-E*01:01 has arginine (Arg). This difference is located outside the peptide-binding groove and does not affect its overall structure [4,5,12]. Consequently, both alleles have a similar range of peptide-binding capabilities, although differences in cell surface stability and peptide affinity have been reported [9], which may influence disease associations. Table 1 summarizes the key features of these two alleles.

HLA-E mRNA is present in all nucleated cells and is at least four times more abundant in the endoplasmic reticulum (ER)**.** However, HLA-E molecules are expressed at low levels on the cell surface and are rapidly internalized after reaching the cell surface [13]. They are subsequently recycled in endosomes unless inflammatory signals like interferons are produced [14]. Recent advances in predictive peptide-binding motifs analysis for pathogens and tumors have verified that hydrophobic amino acids at positions 9 and 2 serve as essential anchors within the HLA-E peptide-binding groove [15,16,17]. The conventional function of HLA-E in regulating the innate immune response has been indicated by substantial research. Its primary function is to present a highly conserved nonameric signal peptide, VMAPRTLVL (VL9-peptides), derived from the leader sequences of classical HLA class Ia molecules, including HLA-A, -B, -C, and -G [18]. HLA-E does not have a VL9 peptide itself. This presentation of this peptide inhibits NK cells by interacting with the inhibitory receptor NKG2A-CD94. A growing body of research has recently demonstrated that various pathogen-derived peptides presented by HLA-E molecules can elicit HLA-E-restricted cytotoxic T-cell responses that capable of directly killing infected cells [19,20,21,22,23,24]. 

**Table 1 cells-14-01983-t001:** HLA-E*01:01 vs. HLA-E*01:03: Expression, Stability, and Disease Associations.

Feature	HLA-E*01:01	HLA-E*01:03	Reference
Amino Acid Position	Arginine (Arg) at position 107 in the α2 domain	Glycine (Gly) at position 107 in the α2 domain	[9,25]
Cell Surface Expression	Lower	Higher	[25,26]
Peptide Binding Affinity	For canonical VL9 peptides, Lower peptide-binding affinity For non-VL9 pathogen-derived peptides, more flexibility for alternative peptides under stress or infection	For canonical VL9 peptides, higher peptide-binding affinity For non-VL9 pathogen-derived peptides, more conserved but stabilize better	[15]
Disease Associations	Viral: Protective against EBV-positive classical Hodgkin lymphoma; lower risk of chronic HCV; protective in BK virus nephropathy	Viral: Increased risk of chronic HCV; associated with HIV protection; linked to CMV reactivation, maintain immune responses post-Ad5-nCoV-vaccination. Associated with severe COVID-19.	[12,26,27,28,29]
	Bacterial: Risk factor for severe bacterial infections	Bacterial: Protective against severe bacterial infections	

During CMV infection, UL40 leader peptides containing the VL9 motif—along with other signal peptide ligands—bind HLA-E directly in the ER via a TAP-independent pathway. This occurs because VL9 is co-translationally inserted and cleaved in the ER, bypassing cytosolic transport [30,31,32]. In contrast, most pathogen- or cytosol-derived peptides require proteasomal processing and TAP-mediated entry into the ER, making them TAP-dependent. However, alternative loading routes have been reported in cells lacking classical MHC-I or TAP function [33,34]. 

HLA-E–restricted T-cell responses to *Mycobacterium tuberculosis* (Mtb) show a dual phenotype, with both protective and tolerogenic effects. Structural and in vitro functional studies demonstrate that HLA-E can present Mtb-derived peptides to cytolytic CD8^+^ T cells, enabling recognition and elimination of infected macrophages, as a protective role [35]. This concept is further supported by non-human primate (NHP) models: CMV vector–based vaccination (RhCMV/TB) induces unconventional Mamu-E–restricted responses that significantly reduce TB disease burden following challenge—by up to 68% in some studies—providing strong in vivo proof-of-principle for MHC-E-targeted vaccine efficacy [36]. Yet, human ex vivo analyses reveal that certain HLA-E–restricted CD8^+^ subsets produce atypical cytokines, including IL-4 and IL-13, as well as mixed cytolytic and suppressive characteristics [37]. These findings raise the possibility that, under specific conditions, HLA-E–restricted responses may be regulatory or heterogeneous rather than strictly protective. This functional diversity likely reflects differences in epitope specificity, TCR repertoire, costimulatory context, and tissue microenvironment.

Extensive in vitro evidence—including peptide-binding assays, crystallographic analyses of HLA-E–peptide complexes, and CD8^+^ T-cell cytotoxicity assays—clearly demonstrates that HLA-E can present pathogen-derived antigens and trigger functional immune responses [16]. Nonetheless, in vivo validation remains limited. Data from RhCMV vaccine studies indicate that Mamu-E–restricted responses are associated with durable, unconventional CD8^+^ T-cell immunity and partial protection against both SIV and TB [36]. Despite this promise, translation to humans remains uncertain. The RhCMV 68-1 strain carries unique genetic deletions and tropism features that drive its atypical restriction profile, and CMV vectors may not replicate these immunological properties without deliberate genetic engineering [38]. Moreover, species-specific differences in MHC-E sequence, splicing isoforms, and CMV immune-evasion genes complicate direct extrapolation from NHP models to human systems. 

Ongoing key conflicts remain, as TAP dependence is not a straightforward binary issue; discrepancies among studies often depend on ligand source and experimental methods. HLA-E–restricted responses in *M. tuberculosis* exhibit both protective and tolerogenic features. In vitro cytotoxicity and structural data support a protective role, whereas ex vivo human studies reveal type-2 and suppressive cytokine profiles. The functional outcome appears highly context-dependent, shaped by the presented epitope and the host immune environment [35]. 

Detailed mechanisms of peptide binding and loading have been extensively reviewed in [25]. This review highlights the current gaps in understanding of T-cell recognition of HLA-E–presented peptides across infections and explores how advancing this gap could guide the development of future HLA-E–based vaccines and immunotherapies targeting microbial pathogens.

## 3. Activation and Inhibition in NK and T Cell Responses

The HLA-E–VL9 complex influences both innate and adaptive immunity by interacting with activating and inhibitory receptors on natural killer (NK) cells and T cells (Figure 1). Pathogens can exploit these receptor–ligand interactions to escape immune detection or enhance immune defense mechanisms.

### 3.1. CD94–NKG2A/C-Mediated Immune Regulation (Innate Receptor)

HLA-E was initially identified as a ligand for the CD94/NKG2A or CD94/NKG2C heterodimeric co-receptor complexes, which are expressed on NK cells [18,39]. CD94/NKG2A functions as an inhibitory receptor, enabling NK cells to distinguish between self and non-self by recognizing nonameric peptides presented by HLA-E. 

Conversely, CD94/NKG2C activates NK cells upon recognizing the same HLA-E complexes. Both co-receptors bind HLA-E/VL9-peptide complexes to either inhibit or activate NK cell cytotoxicity (Figure 1b,c). Importantly, CD94/NKG2A receptor exhibits higher binding affinity for HLA-E/VL9-peptide complexes compared to CD94/NKG2C, although both receptors recognize the HLA-E/VL9-peptide through a similar structural motif [40,41]. 

Expression levels differ significantly between these receptors: CD94/NKG2A is expressed on around 50% of NK cells, 5% of CD8^+^ T cells, and from 20% to 90% of the specific Vδ2 T cell subset in peripheral blood of healthy humans, whereas CD94/NKG2C is expressed less frequently on these cells [42,43]. This quantitative difference explains why inhibitory signaling generally predominates under homeostatic conditions. Blocking the CD94/NKG2A could therefore activate the Vδ2 T cell subset. Moreover, blocking CD94/NKG2A enhances the cytotoxic activity of NK cells against infected and cancer cells that overexpress HLA-E [44]. Polymorphisms in HLA class I signal peptides, particularly in the VL9 sequence presented by HLA-E, further affect this activation–inhibition balance by modulating CD94/NKG2 receptor interactions.

### 3.2. TCR-Mediated Immune Response (Adaptive Receptor)

T cell receptors (TCRs) on CD8^+^ T cells can recognize both VL9 and non-VL9 peptides presented by HLA-E. Many pathogens downregulate the expression of classical HLA-Ia molecules to evade recognition by CD8^+^ T cells; consequently, non-classical HLA-I molecules present peptides that trigger the cellular immune system to detect ‘missing self’ abnormalities. Figure 1a shows how TCR binding to HLA-E–peptide complexes triggers effector functions, such as secretion in IFN-γ/TNF-α and perforin/granzyme-mediated cytotoxicity. Over the past two decades, an increasing number of studies have reported pathogen-specific HLA-E-restricted CD8^+^ T cells in response to intracellular infections, including bacterial and viral pathogens. However, although these analyses highlight the vital role of these CD8^+^ T cells in host defense, they have not yet characterized these cells in detail [45,46,47]. For example, during *M. tuberculosis* infection, HLA-E acts as an essential mediator for activating CD8^+^ T cells by presenting multiple mycobacterial peptides to the TCR [22,48]. Similarly, studies in CMV-seropositive individuals show that a proportion of CD8^+^ T cells is HLA-E restricted [46,49,50,51]. Another study demonstrated that CD8^+^ T cells can recognize HIV-Gag peptides bound to HLA-E and suppress HIV-1 replication in vitro [24]. 

In SARS-CoV-2 infection, the SP1 peptide derived from Spike stabilizes HLA-E expression on lung epithelial cells [52]. In patients with severe to moderate COVID-19, HLA-E is upregulated in the lung epithelium [53]. Hammer et al. reported that a SARS-CoV-2 peptide from the non-structural protein (Nsp) 13 can be presented by HLA-E, activating NKG2A^+^ NK cells in individuals with COVID-19, and restricting SARS-CoV-2 replication in infected lung epithelial cells in vitro [54]. Genetic variations in the NKG2C/HLA-E axis also influence disease severity [55]. A key study into HLA-E–restricted CD8^+^ T cell responses in individuals recovering from SARS-CoV-2 infection identified five viral peptides capable of eliciting HLA-E-restricted CD8^+^ T-cell responses [56].

## 4. HLA-E in NK and T Cell Response to Viral Infection

Viruses reside within host cells because they rely on human hosts to survive. Their proteins are processed and presented by HLA class I molecules—including the non-classical HLA class I—to CD8^+^ cytotoxic T lymphocytes (CTLs), which recognize HLA class I/peptide complexes via TCR to mediate antiviral responses. Across multiple viral infections (CMV, SARS-CoV-2, HBV, and HIV), HLA-E acts as a central immune modulator, balancing NK cell inhibition and CD8^+^ T-cell activation. These viruses exploit HLA-E to evade classical CD8^+^ T cell responses by mimicking self-peptides or altering peptide presentation. In contrast, specific host responses leverage HLA-E’s limited polymorphism to mount broad antiviral HLA-E-restricted T cell responses [57]. 

### 4.1. HLA-E and CMV

CMV is a double-stranded DNA virus classified within the herpesvirus family that [58] downregulates classical HLA class Ia molecules during infection to escape immune detection. Reduced expression of HLA class I molecules limits the availability of leader sequences for interaction with HLA-E, thereby increasing NK cell-mediated lysis. However, to evade immune detection by NK cells, CMV uses the viral UL40 peptide to mimic host HLA class I leader sequences, stabilizing HLA-E and preventing NK-cell killing via NKG2A/CD94 interaction [39]. This mechanism enables viral evasion from NK cells. Subsequently, expansion of NKG2C^+^ NK cells with adaptive features—such as epigenetic remodeling, enhanced antibody-dependent cellular cytotoxicity (ADCC)—contributes to viral control [40,59]. Studies on adaptive NK cells have identified three unconventional HLA-E-restricted 15-mer peptides derived from the CMV pp65 protein that elicit NK cell memory responses specific to CMV [60]. Thus, the NKG2A/NKG2C–HLA-E axis represents a critical checkpoint balancing NK cell inhibition and activation in both homeostasis and infection.

CMV UL40 peptide and any other CMV peptides with sequence homology to the leader sequences of HLA class Ia molecules are unlikely to activate HLA-E-restricted CD8^+^ T cells since these peptides will be identified as “self,” thus preventing initiation of a CD8^+^ T cell response. However, those individuals with HLA class Ia molecules that possess leader sequences differing from the CMV UL40 peptides generate a strong CD8^+^ T cell response, particularly in individuals lacking HLA-C alleles with identical leader sequences or in individuals who develop a specific TCR type (TRBV14) that has been found to recognize the HLA-E/VMAPRTLIL peptide complex [31,61]. HLA-E/UL40-restricted CD8^+^ T cells displayed an effector-memory phenotype and were detected in approximately 28.7% of CMV-positive kidney transplant recipients and 32.0% of CMV-positive healthy donors. Notably, many of these HLA-E/UL40-restricted CD8^+^ T cells were cross-reactive: they recognized not only the UL40 peptide derived from the infecting CMV strain, but also non-canonical UL40 variants and autologous HLA-I signal peptides with sequence homology [62]. 

Unconventional HLA-E-restricted CD8^+^ T cells are a significant component of long-term CMV immunity, approximately the same frequency as conventional CD8^+^ T cells. HLA-E/UL40 is present in nearly one-third (27.5%) of seropositive healthy adults. Although HLA-E-restricted CD8^+^ T cells show lower TCR avidity than HLA-A2-restricted ones, they have similar CD3/CD8 co-receptor expression [63]. A recent study has highlighted an unconventional subset of HLA-E-restricted CD8^+^ T cells, in addition to the classical HLA class I-restricted CD8^+^ T cell responses towards pp65 and IE1, which represent up to 38% of circulating CD8^+^ T cells in human CMV-infected hosts. Therefore, individuals lacking matching leader sequences can mount strong HLA-E–restricted CD8^+^ T-cell responses [64]. 

Soluble HLA-E (sHLA-E) shows an immunoregulatory role during cytomegalovirus (CMV) infection by binding to the inhibitory receptor CD94/NKG2A on NK cells, thereby evading responses. Elevated sHLA-E levels were strongly associated with subsequent high-level CMV replication in lung transplant recipients. Mechanistically, sHLA-E concentrations as low as 0.05 ng/mL were sufficient to inhibit NKG2A^+^ NK cell activity. Therefore, sHLA-E could be considered a potential biomarker for predicting CMV reactivation risk and guiding personalized immunomonitoring, though clinical validation remains limited. An important consideration is whether sHLA-E reflects general inflammation or virus immune evasion. Current evidence indicates that sHLA-E release occurs in both systemic inflammatory states and virus/tumor-specific contexts. For example, increasing soluble HLA-E contributes to immune escape in gastric cancer (GC) cell lines. Tumor cells not only upregulated the expression of surface HLA-E but also a soluble form that can circulate in the environment and potentially bind NKG2A on NK cells and suppress their activity. Therefore, sHLA-E could be considered a biomarker of disease severity in tumors. Considering the significant release of sHLA-E by infected/tumor tissues and its up-regulation by inflammatory cytokines, it is important to explore the potential role of sHLA-E in immune responses to tumors and viral infections [65,66,67,68].

### 4.2. HLA-E and SARS-CoV-2

SARS-CoV-2 is a single-stranded, enveloped RNA virus [69] that induces upregulation of HLA-E on lung epithelial cells and transfected cell lines expressing viral proteins such as Spike and NSP13. 

T cells play a crucial role in the immune response to COVID-19. Since the start of the pandemic, studies have revealed a combination of immune evasion and increased transmissibility in several SARS-CoV-2 variants, including Alpha (B.1.1.7), Delta (B.1.617.2), and Omicron (B.1.1.529) strains [70,71,72,73]. The continuous emergence of mutated variants and their capacity to evade immune defenses has raised a significant threat to public health and the global economy. Several HLA class II alleles, including HLA-DRB1*04, HLA-DRB1*08:02, and HLA-DRB1*09:01, have been associated with the disease severity, suggesting that HLA class II molecules play a role in immune modulation [74,75,76,77,78]. Additionally, HLA class I alleles such as HLA-C*04:01 may also affect clinical outcomes and epidemic patterns of SARS-CoV-2, with certain alleles linked to severe cases [78].

SARS-CoV-2 employs multiple strategies to evade NK cell-mediated immunity. Downregulation of ligands by the Nsp1 peptide for the activating NK receptor NKG2D (NKG2D-L) reduces NK cell recognition, while modulation of HLA-E/NKG2A signaling suppresses NK cytotoxicity [52,79]. In severe COVID-19, NK cells exhibit an exhausted phenotype, exhibiting poor cytotoxic function and cytokine production upon stimulation [80]. Interestingly, a viral peptide from Nsp13 (Nsp13_232–240_) disrupts this inhibitory pathway by preventing NKG2A binding, enabling NKG2A^+^ NK cells to regain activity and restrict viral replication in infected lung epithelial cells in vitro [54]. Spike peptide presentation by HLA-E further shapes NK responses by inhibiting NKG2A^+^ NK cells while activating NKG2C^+^ NK cells, which display adaptive, memory-like features. This dual effect suppresses certain NK cell subsets while enhancing others, influencing the overall antiviral response.

A key study on HLA-E-restricted CD8^+^ T cell responses in convalescent patients with COVID-19 disease identified five SARS-CoV-2 peptides that elicit HLA-E-restricted CD8^+^ T-cell responses at similar frequencies as classical HLA class Ia-restricted CD8^+^ T cells. Notably, the generated HLA-E–restricted peptide-specific T cell clones, with diverse TCRs, demonstrated the ability to suppress SARS-CoV-2 replication in Calu-3 human lung epithelial cells. In contrast, the downregulation of classical HLA class I molecules occurs upon infection in Calu-3 cells. These findings suggest that HLA-E–restricted T-cell responses may contribute to viral control and support the idea that HLA-E–based vaccines targeting conserved epitopes could offer broad, variant-independent protection [23,57].

### 4.3. HLA-E and HBV

HBV is a double-stranded DNA virus belonging to the Hepadnaviridae family [81]. In HBV infection, HLA-E expression on hepatocytes plays a critical checkpoint for NK cell regulation. For example, in patients with acute liver failure (ALF) caused by HBV, Differentially Expressed Genes (DEGs)—including HLA-E and pathways related to NK cell-mediated cytotoxicity and antigen presentation—are significantly upregulated compared to healthy controls. Therefore, HLA-E may influence disease outcomes by regulating NK cell activity [82]. In chronic HBV (CHB) patients, soluble HLA-E is significantly higher than in healthy controls and causes inhibition of NK-cell cytotoxicity [83].

Adaptive NK cells, characterized by high NKG2C expression, recognize HLA-E and exhibit enhanced effector functions, including cytotoxicity and antibody-dependent cellular cytotoxicity (ADCC). Unlike conventional NK cells, which often become dysfunctional in chronic HBV, NKG2C^+^ adaptive NK cells maintain robust IFN-γ production and killing capacity, correlating with better viral control [84].

Acute symptomatic hepatitis B is primarily controlled by HBV-specific CD4^+^ and CD8^+^ T-cell responses, as well as neutralizing antibodies. HBV-specific CD8^+^ T cells directly destroy infected hepatocytes and produce cytokines such as IFN-γ to eliminate the virus and recruit other immune cells. In chronic patients, the frequency of HBV-specific CD8^+^ T cells is considerably reduced [85,86]. Furthermore, because of the liver’s tolerogenic properties, co-stimulatory molecules are low, and co-inhibitory ligands PD-L1/2 are upregulated, which eventually induces T cell tolerance rather than activation [87]. Another evasion of immune response by the virus is that classical T-cell responses to HBV are also influenced by genetic variation in the HLA class I profile and the presence of different HBV subtypes across populations. Therefore, each viral epitope, from nucleocapsid, envelope, polymerase, and X proteins, can elicit T cell responses differently and vary significantly between individuals and ethnic groups [87]. 

HLA-E expression influences viral persistence or clearance, thereby modulating disease outcomes. For example, functional polymorphisms in HLA-E*01:01/01:03 (rs1264457) and HLA-G polymorphisms (rs41551813 and rs1130355) influence the outcome of HBV infection; the A allele of HLA-E (rs1264457) correlates with a significantly increased chance of spontaneous HBV clearance. In contrast, elevated sHLA-E levels were associated with HBV persistence and chronic infection [88]. HBV antigens (RhCMV/HBV) expressed by 68–1 rhesus cytomegalovirus vaccine vectors can induce MHC-E-restricted CD8^+^ T cells in rhesus macaques (RM) [89]. A recent study identified the HBV Env371–379 (L6I) peptide as the first verified HLA-E–restricted epitope, capable of inducing HBV-specific CD8^+^ T cells in both infected and naive individuals after they were expanded in vitro using L6I peptide/HLA-E complexes [90].

### 4.4. HLA-E and HIV

HIV is a lentivirus with single-stranded RNA that belongs to the Retroviridae family [91]. In HIV-1 infection, HLA-E functions as a key immunoregulatory ligand by presenting peptides to HLA-E–restricted CD8^+^ T cells and engaging the inhibitory receptor NKG2A/CD94 on NK cells. 

HIV evades NK cell-mediated cytotoxicity by upregulating HLA-E expression on infected T cells, using the capsid protein p24_14–22_ peptide to stabilize HLA-E and suppress NK cell responses [92]. KIR2DL1, an inhibitory receptor on NK cells, binds to HLA-C group 2 alleles on target cells and reduces NK cell cytotoxicity. Therefore, KIR2DL1^–^ NK cells enable stronger antiviral responses. Interestingly, although HLA-E presents the conserved Gag-derived peptide AISPRTLNA during HIV infection, this peptide does not efficiently engage NKG2A/CD94, limiting inhibitory signaling. Consequently, KIR2DL1-negative, NKG2A-positive NK cells can remain cytotoxic despite HLA-E expression [93].

Primary HIV-1 strains can downregulate HLA-E on infected CD4^+^ T cells via Nef, reducing recognition by HLA-E–restricted CD8^+^ T cells [94]. Proteasome inhibition further reduces HLA-E expression, which boosts NK cell cytotoxicity by preventing NKG2A engagement and may help reverse HIV latency. In vitro and ex vivo results suggest that disrupting the NKG2A/HLA-E axis could enhance “shock-and-kill” strategies for reactivated cells by increasing NK cell-mediated clearance [95]. A specific subset of NK cells expressing NKG2C receptors, called adaptive NK cells, recognize infected cells via an HLA-E molecule presenting certain HIV peptide fragments [96]. Although adaptive NKG2C^+^ NK cells expand in HIV-infected individuals, this phenomenon is primarily driven by coexisting CMV infection rather than HIV-derived HLA-E ligands, as HIV peptides bind HLA-E weakly and inconsistently. 

However, contradictory findings exist regarding HIV peptide binding to HLA-E. Some peptides strongly stabilize HLA-E, while others bind weakly or inconsistently. Experiments on virus inhibition demonstrated that TCR mediated the interaction between the RL9 peptide and HLA-E, and that cells infected with HIV-1 displayed the RL9-HIV-1 peptide-HLA-E complex. However, the HIV-derived RL9 peptide, weakly stabilizes the HLA-E molecule due to a large aromatic residue at P3, which disrupts the peptide’s conformation and decreases the stability of the HLA-E-RL9 complex [16]. 

One promising approach is to focus on antigen presentation by the nonpolymorphic HLA-E molecule. Over 400 HIV-1-derived 15-mer peptides were screened and analyzed for the cell-surface stabilization of HLA-E01:01 and HLA-E01:03 molecules upon peptide binding. Four novel 9-mer peptides (PM9, RL9, RV9, and TP9) derived from the 15-mer binders specifically stabilized the surface expression of HLA-E*01:03, and five other peptides (EI9, MD9, NR9, QF9, and YG9) showed binding signals. A crystal structural study examining HLA-E in complex with HIV-derived peptides and its interaction with CD94/NKG2 receptors revealed that HLA-E exhibits a broad tolerance for hydrophobic and polar residues within its primary binding pockets, highlighting its adaptability in presenting pathogen-derived antigens. This comparative analysis explains why some studies show inconsistent presentation of HIV-derived peptides and highlights the need for standardized binding assays to accurately evaluate immunogenic potential [96].

CD8^+^ T cells are essential for controlling HIV-1 infection by recognizing viral peptides presented by HLA class I molecules; however, CD8^+^ T cells show a wide range of inhibitory capacity against infection and viral load among individuals [97]. Similar to the other viral diseases, HLA class I-associated HIV-1 amino acid polymorphisms lead to immune escape from HLA class I-restricted CD8^+^ T cells, either through loss of peptide recognition or through mutations that maintain peptide recognition but result in ineffective T cell responses and dysfunctional T cell phenotypes [98,99]. These mechanisms enable HIV-1 to evade the immune pressure exerted by HLA class I-restricted CD8^+^ T cells, a phenomenon referred to as “viral adaptation” [100]. Consequently, despite over four decades of research, developing a vaccine for HIV-1 has been challenging due to genetic diversity and the evasion of detection and clearance by the immune system of the host [93,99]. 

HLA-E-restricted CD8^+^ T cells, which can recognize and respond to *M. tuberculosis*, might provide protective immunity even in the presence of HIV-1. The presence of both pathogens led to a decrease in HLA-A2 expression on macrophages, resulting in reduced effectiveness of HLA-A2-restricted CD8^+^ T cells in killing infected cells and controlling the growth of intracellular *M. tuberculosis*. In contrast, the expression of HLA-E and the activity of HLA-E-restricted CD8^+^ T cells were not negatively affected by HIV-1 [22].

CD8^+^ T cells responding to two other epitopes, which had been previously associated with protection in an SIV/rhesus macaque model: a newly identified subdominant Gag-KL9 and a well-described immunodominant Gag-KF11 [99]. Yang et al. [24] demonstrated that CD8^+^ T cells, which recognize a Gag peptide presented by HLA-E, are capable of suppressing HIV-1 replication in vitro. To confirm that this antiviral activity was specifically mediated by the HLA-E–restricted T cell receptor, they cloned these HLA-E-restricted HIV-1-specific CD8^+^ T cells and transduced them with their TCRs. This process effectively suppressed HIV-1 replication in CD4^+^ T cells in vitro.

In contrast, in a recent study on HLA-E-restricted peptide drive form the HIV virus, using immunopeptidomic and bioinformatic approaches, HIV-infected cells do not reliably present peptides via HLA-E. Previously reported HLA-E ligand, Gag_275–283_ peptide, also displayed inconsistent presentation in killing assays, requiring continuous loading of HLA-E molecule with exogenous peptide to elicit consistent T-cell responses [98]. 

Discrepancies across studies analyzing HIV peptide binding to HLA-E mainly arise from peptide variability. Strong binders usually have optimal anchor residues in the HLA-E pockets, such as hydrophobic residues at P2 and P9, while weak binders contain bulky or polar residues that destabilize the complex (e.g., AISPRTLNA, RL9), leading to poor stabilization of HLA-E. Other factors, like experimental conditions, should also be considered; some assays depend on exogenous peptide loading, which can artificially boost binding signals. 

## 5. HLA-E in NK and T Cell Response to Bacterial Infection

CD8^+^ T cells recognize endogenous antigens from intracellular bacteria presented by HLA class I molecules and lyse infected target cells, thereby playing a crucial role in the immune response. Notably, in the absence of HLA class I-restricted CD8^+^ T cells, HLA-E-restricted CD8^+^ T cells that are capable of recognizing pathogen-derived peptides have been documented for *M. tuberculosis* and *Salmonella* (*S.*) Typhi.

### 5.1. HLA-E and M. tuberculosis

HLA-E plays a pivotal role in coordinating innate and adaptive immune responses during bacterial infections by functioning both as a ligand for NK-cell receptors and as a non-classical antigen-presenting molecule for CD8^+^ T cells [45].

Analysis of *M. tuberculosis*-infected human dendritic cells shows that HLA-E is preferentially enriched in phagosomes compared to classical HLA-A2, with abundant HLA-E/peptide complexes detected in these compartments [101]. Screening of the Mtb genome identified multiple HLA-E–binding peptides capable of inducing HLA-E-specific CD8^+^ T cell responses [35]. Mass spectrometry further revealed 28 immunogenic HLA-E–restricted peptides from 13 Mtb proteins, with Rv0634A_19–29_ eliciting strong IFN-γ responses in most donors [102]. Moreover, a peptide derived from the *M. tuberculosis* protein MPT32, which requires N-terminal O-linked mannosylation for recognition, underscores the importance of post-translational modifications in immune recognition. This finding is particularly noteworthy as it is the first report detailing a modified *M. tuberculosis*-derived protein antigen that elicits an HLA-E-specific CD8^+^ T cell response [103]. 

NK cells provide an early innate defense against *M. tuberculosis* by recognizing and killing infected cells. Recent studies show that NK cell subsets undergo significant changes during TB infection: CD8α^+^ NK cells, a mature and highly functional population, are associated with resistance to primary infection but are progressively lost during latent TB and active disease. While NK cell interaction with HLA-E through NKG2A/C receptors may contribute to immune surveillance, no significant differences in NKG2C expression were observed between latent and active TB groups [104]. Interestingly, HLA-E/NKG2C receptor expression is significantly higher in patients who have completed their treatment than in donors undergoing active anti-tuberculosis therapy [105].

In the adaptive immune response, activated CD8^+^ and CD4^+^ T cells migrate to the lungs to target infected tissue cells; their ability to control the infection is limited due to *M. tuberculosis* strategies that evade detection and induce T cell exhaustion. *M. tuberculosis*-infected cells evade classical HLA-I-restricted T cell recognition; therefore, it is important to identify antigens less impacted by evasion mechanisms, such as the highly conserved antigens of *M. tuberculosis* that HLA-E can present and elicit strong T cell responses [106]. 

HLA-E-restricted CD8^+^ T cells specific for *M. tuberculosis*-derived peptides are detectable in the peripheral blood of individuals with either active tuberculosis or latent *M. tuberculosis* infection [107,108]. These cells display multifunctional phenotypes, combining cytolytic activity, IFN-γ and TNF-⍺ secretion, and regulatory functions mediated partly by membrane-bound TGF-β [35,109], which serve to protect across diverse populations. HLA-E-restricted CD8^+^ T cells represent a significant subset of effector cells involved in the immune defense during active *M. tuberculosis* infection. The frequency of these cells is significantly increased in the circulation of aTB patients, and their presence in patients with and without HIV co-infection underscores their potential contribution to protective immunity in diverse populations [48].

In a comprehensive vaccination study, BCG vaccination elicits only limited expansion of Mtb-specific HLA-E-restricted T cells in humans and non-human primates. In contrast, natural infection induces robust activation, underscoring the differential immunogenicity of BCG versus natural exposure in shaping HLA-E-mediated immunity [110]. 

More recently, several novel *M. tuberculosis* peptides capable of binding HLA-E were identified using an optimized prediction algorithm for the identification of pathogen- or self-peptides [111]. These HLA-E/epitopes were recognized not only by CD8^+^ T cells but also by CD4^+^ T cells in humans and rhesus macaques. HIV co-infection further changes the memory distribution within these subsets. Phenotypic analysis reveals that HLA-E-restricted T cells encompass diverse memory subsets and express exhaustion markers at levels similar to the overall T cell population. The presence of HLA-E/Mtb-specific T cells in people with active tuberculosis (aTB) and tuberculous infection (TBI) emphasizes the possibility of using HLA-E as a target for TB vaccination [37].

Structural analyses reveal that HLA-E exhibits broad peptide-binding flexibility, tolerating both hydrophobic and polar residues at key anchor positions. For example, the P2 residue of the *M. tuberculosis* peptide (p44) fits well in the B pocket of HLA-E, and substitutions with residues such as glutamine or phenylalanine are well tolerated without inducing major conformational shifts. Furthermore, even though C-terminal substitutions such as P9-Phe present spatial constraints, HLA-E can still accommodate these variations, illustrating its versatility and potential alternative antigen-presenting functions in vivo [16,112]. This versatility enables recognition of diverse Mtb peptides and positions HLA-E–restricted CD8^+^ T cells as a distinct immune subset bridging innate and adaptive responses. These cells represent promising targets for vaccine design and immunotherapy against tuberculosis.

### 5.2. HLA-E and S. Typhi

*S. Typhi* is a rod-shaped, gram-negative bacterium of the Enterobacteriaceae family. Multifunctional CD8^+^ cytotoxic T lymphocytes are essential for preventing the onset of typhoid disease and providing protection after the Ty21a vaccine [113], particularly those cells that target the *S. Typhi* antigen presented by HLA-E.

Nonamer-derived *S. Typhi* peptides were identified for HLA-E-specific recognition in the PBMC of Ty21a typhoid vaccine recipients. Binding of these bacterial-derived peptides to HLA-E activates CD8^+^ T cells, enhancing cytotoxic activity and IFN-γ production which can effectively eliminate *S. Typhi*-infected cells [114]. HLA-E-restricted CD8^+^ T cell responses targeting *S. Typhi* are detectable both prior to and following vaccination with the Ty21a strain of typhoid. This finding highlights the importance of long-lasting, multifunctional HLA-E-restricted CD8^+^ responses in adaptive immunity to *S. Typhi*, marking the first demonstration of such cells after immunization in either animals or humans [115].

HLA-E–restricted T cells are associated with delayed disease onset and protection. However, limited pediatric data suggest that younger children may have a reduced capacity to mount protective cell-mediated immune responses post-vaccination. Mass cytometry analyses and tSNE (t-distributed Stochastic Neighbor Embedding) analysis revealed age-dependent differences in the abundance and function of gut-homing effector memory (T_EM_) and terminally differentiated (T_EMRA_) T cell subsets. The functional features of HLA-E-restricted CD8^+^ T_EM_ cells demonstrate elevated levels of MIP1β, CD107a, TNF-α, IL-2, IL-17A, IFN-γ, and Granzyme B following Ty21a vaccination. This increase occurs in response to HLA-E-restricted S. Typhi antigen presentation, compared to responses prior to vaccination. Maturation of HLA-E–restricted immunity highlights its importance in improving pediatric vaccine strategies [116].

A representative list of known HLA-E–restricted epitopes derived from viral and bacterial pathogens is presented in Table 2.

## 6. HLA-E in Immunotherapy and Vaccine Development for Infectious Diseases

The HLA-E molecule has traditionally been recognized for its role in regulating NK cell activity through interactions with the inhibitory receptor NKG2A. However, recent research has expanded its functional relevance to the adaptive immune system, particularly in vaccine design and T cell immunotherapy for infectious diseases, given the limited display of polymorphisms (Figure 2).

### 6.1. HLA-E–Based Vaccine and TCR Strategies for Tuberculosis

The limited efficacy of the current BCG vaccine highlights the urgent need for new tuberculosis (TB) vaccines. HLA-E–restricted, M. tuberculosis–specific CD8^+^ T cells can control intracellular bacterial growth, making HLA-E a promising target for novel vaccine strategies. Although donor-unrestricted and HLA-E–restricted T cell responses show promise, identifying pathogen-derived HLA-E–binding peptides remains challenging because of limited predictive tools.

One study developed a UV-mediated HLA-E/peptide binding assay to generate data for training a neural network (NNAlign), which created HLA-E binding motifs. These motifs were used to screen *M. tuberculosis* genomes for potential peptides, which were synthesized and validated through iterative binding assays. This approach improved peptide prediction accuracy and led to the discovery of novel *M. tuberculosis* peptides capable of activating CD8^+^ T cells in exposed humans [111]. Recently, our group demonstrated that delivering HLA-E–restricted *M. tuberculosis* peptides via a modified Bordetella pertussis toxin (CyaA toxoid) significantly expanded CD8^+^ T cells inducing a predominant Tc1 cytokine profile with a significant increase in IFN-γ and exhibiting cytotoxicity against infected cells [21], showing that CyaA toxoids shift the polarization of the immune response from a typical Th2 type to a mixed Th1/Th2 type of response [119].

Paterson et al. developed a bispecific TCR-based molecule that specifically recognizes HLA-E bound to an *M. tuberculosis* inhA–derived peptide. This affinity-enhanced TCR, linked to an anti-CD3 domain, effectively induced T cell–mediated killing of infected macrophages. Owing to the low polymorphism of HLA-E, this donor-unrestricted TCR-based immunotherapy could broadly target TB infections and overcome limitations of conventional TCR therapies [120].

### 6.2. CMV-Vectored Vaccines for SIV

SIV and HIV rapidly evade host immunity, making early immune control crucial for effective intervention. Hansen et al. showed that rhesus CMV (RhCMV) vectors encoding SIV antigens generate long-lasting, high-frequency effector-memory CD8^+^ T-cell responses that enable early control and, in some cases, clearance of SIVmac239 infection [56]. These responses persisted even after CD4^+^/CD8^+^ depletion, underscoring the robustness of CMV-based immunity.

RhCMV/SIV vectors also elicit unconventional MHC-E–restricted CD8^+^ T cells that are critical for protection. Disruption of CMV immunomodulatory genes enhances these responses, whereas restoring those genes eliminates protection [51]. The Rh67-derived VL9 peptide facilitates MHC-E trafficking and primes these CD8^+^ T cells; deleting Rh67 shifts immunity toward MHC-II restriction and abolishes protection [121]. Vaccine efficacy is species-specific as well. In cynomolgus macaques, only matched CyCMV/SIV vectors generated MHC-E–restricted CD8^+^ T cells and achieved SIV control in roughly half of the animals, with IL-15–associated transcriptional signatures paralleling those seen in protected rhesus macaques [51]. Picker et al. [122] highlighted CMV’s advantages as a vaccine vector, particularly its capacity to induce broad and unconventional CD8^+^ T-cell responses through MHC-Ia, MHC-II, and MHC-E pathways. Among these, MHC-E–restricted responses uniquely conferred stringent SIV control, pointing to their potential relevance for HIV-1 vaccine strategies.

Malouli et al. [123,124] identified the CMV UL18 gene as a key immune evasion factor that blocks HLA-E–restricted T-cell priming via LIR-1 binding. Disrupting UL18–LIR-1 interactions restored unconventional responses, a vital step for developing CMV-based HIV vaccines. Recent CMV vector innovations underscore the central importance of MHC-E–restricted immunity. The canonical RhCMV 68-1 vector elicits effector-memory CD8^+^ T cells targeting SIV via MHC-E and achieves complete viral suppression in roughly 60% of macaques [125]. Updated ΔgL “single-cycle” vectors, designed for improved safety, preserved these protective responses and produced ~70% replication arrest [126]. Moreover, pairing RhCMV/SIV vaccination with sub-protective neutralizing antibodies further boosted control (44% vs. 11%), highlighting the potential of integrated, multimodal HIV vaccine approaches [127].

### 6.3. TCR-Based Therapy for HBV

A recent study engineered T cells lacking HLA class I molecules to avoid recognition by recipient CD8^+^ T cells and incorporated a modified receptor, dtHLA-E4, to reestablish NK-cell tolerance. This optimized construct showed enhanced surface stability and stronger inhibitory activity toward NK cells while limiting engagement by CD8^+^ T cells, thereby lowering immunogenicity and improving T-cell persistence for therapeutic use [128].

Another approach, the Immune Mobilizing Monoclonal T Cell Receptors Against Viruses (ImmTAV^®^) platform (ImmunoCore, UK, Oxford), employs soluble, high-affinity TCRs fused to CD3-engaging domains to redirect polyclonal functional T cells to eliminate virus-infected cells. An HLA-A02:01–restricted ImmTAV molecule targeting an HBV Env-derived peptide effectively lysed HBsAg-positive hepatocytes in vitro [129]. However, given the limited prevalence of HLA-A*02:01 (<30%) among chronic HBV patients, Murugesan et al. [90] focused on HLA-E–restricted CD8^+^ T cells due to the molecule’s high conservation. They identified three variants of the HLA-E–binding Env_371–379_ peptide through bioinformatics and peptide-binding assays and generated an HLA-E–restricted ImmTAV with picomolar affinity that redirected polyclonal T cells to efficiently kill HBV-associated hepatocellular carcinoma cells.

### 6.4. Design Principles of HLA-E–Based Vaccines and Potential Risks of Cross-Reactivity

HLA-E, a nonclassical MHC class I molecule with limited polymorphism, presents both self-derived and pathogen-derived peptides to CD8^+^ T cells and interacts with NK cells via NKG2A/C receptors. This unique dual role provides a translational advantage: it enables the design of broadly applicable vaccines that bypass HLA restriction and elicit donor-unrestricted cytotoxic responses [36,120].

Clinically, these properties are being leveraged to design next-generation vaccines and cell therapies for chronic infections such as Mtb, HIV, and HBV, where traditional HLA-I–restricted strategies have yielded limited success.

The design of HLA-E–based vaccines focuses on selecting conserved, high-affinity pathogen peptides that induce robust, durable CD8^+^ T-cell responses while maintaining NK cell homeostasis. Interestingly, a K562-cell-based line expressing HLA-E, generated by retroviral transduction, showed that it could enhance the adaptive NK cell response with CD57^−^ KIR2DL2/3^+^ NKG2C^+^ NKG2A^−^ phenotype. These K562-21E feeder cells can serve as therapeutic effectors or for studying NK cell maturation following HLA-E peptide presentation [130].

Persistent vectors such as CMV have shown that HLA-E–restricted CD8^+^ T cells can drive near-sterilizing control of SIV in primate models [36,56,122]. Likewise, HLA-E–focused immunotherapies—including engineered TCR bispecifics and ImmTAV^®^ molecules—are progressing toward preclinical and early clinical testing for HIV and HBV [90,120].

Clinical translation must address the risk of cross-reactivity arising from HLA-E’s ability to present both pathogen and self-peptides, particularly VL9 signal sequence-derived peptides from classical HLA-I molecules. Structural similarity between microbial and host epitopes can provoke autoreactive or off-target CD8^+^ T-cell responses, potentially leading to immunopathology. There is growing focus on computational approaches like PepSim and THNet that leverage peptide–HLA structural modeling and machine learning to predict T-cell cross-reactivity. These approaches are enabling researchers to better characterize and predict potential off-target immune reactions, a key step toward developing safer T-cell–based immunotherapies [131,132].

## 7. Conclusions

HLA-E occupies a pivotal position at the interface of innate and adaptive immunity. This non-classical MHC class Ib molecule exhibits minimal polymorphism and is relatively resistant to downregulation by viruses, including HIV, making it an attractive target for vaccines in infectious disease and oncology. CD8^+^ T cell subsets restricted by HLA-E have been documented in CMV, HIV, EBV, HCV, and *Mycobacterium tuberculosis*, often showing substantial antigen-specific expansion. Its peptide repertoire spans canonical leader sequences and pathogen- or stress-derived noncanonical epitopes, enabling broad T cell responses from limited peptide pools. Functional diversity includes cytotoxic, microbicidal, regulatory, and immunomodulatory roles; however, phenotypic exhaustion (e.g., in HIV/TB coinfection) may impair efficacy unless modulated.

Despite these insights, major gaps remain regarding in vivo prevalence, functional biomarkers, and longitudinal contributions to infection control and tumor immunity. Therefore, future studies characterizing functional phenotypes—e.g., cytokine profiles, exhaustion markers, cytotoxic potential, in various clinical contexts are necessary. 

Quantifying HLA-E-restricted CD8^+^ T cell frequencies and characterizing phenotypes—cytokine profiles, exhaustion markers, cytotoxic potential—across clinical contexts are essential to refine its role as a biomarker and accelerate immunotherapy development. 

Future research should integrate advanced methodologies: CRISPR-based models to dissect gene regulation and peptide presentation; single-cell transcriptomics and spatial imaging to map tissue-level interactions; and longitudinal cohorts correlating soluble HLA-E with clinical outcomes in infections such as HBV or CMV. Predictive algorithms for peptide presentation and standardized tetramer reagents will further enable consistent monitoring and translational progress (Figure 3).

Combining HLA-E-based strategies with next-generation immunotherapies, including checkpoint inhibitors and TCR-engineered platforms like ImmTAV^®^, offers transformative potential. ImmTAV^®^ molecules, which redirect diverse T cells toward specific peptide–HLA complexes, could exploit HLA-E’s non-polymorphic nature for universal targeting of intracellular antigens. Finally, vaccination approaches leveraging minimal peptide sets to elicit robust HLA-E-restricted responses, potentially combined with checkpoint modulation (e.g., anti-PD-1) to reverse exhaustion represent a promising frontier. These integrated efforts will bridge fundamental immunology with clinical innovation, positioning HLA-E as a key element of future vaccines and immunotherapies.

## Figures and Tables

**Figure 1 cells-14-01983-f001:**
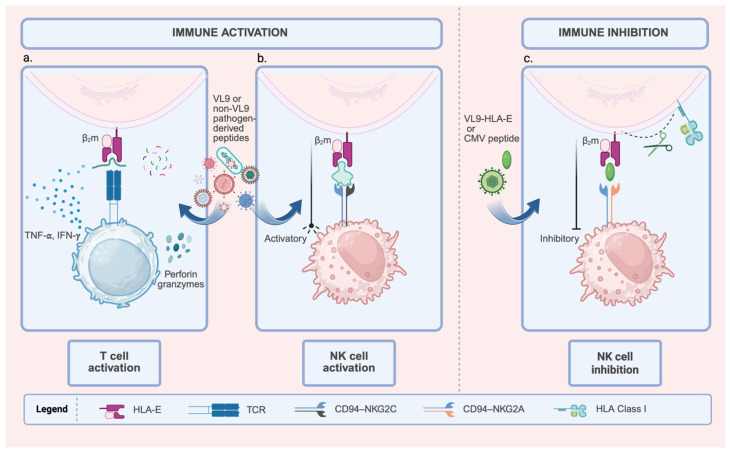
Dual Role of HLA-E in Modulating Immune Responses: Activation and Inhibition. This figure shows the diverse functional outcomes of HLA-E, which depend on the peptide and the immune cell receptor. HLA-E is complexed with β_2_m (Beta-2 microglobulin). (**a**). T cell activation: HLA-E loaded with pathogen-derived peptides (VL9 or non-VL9) is recognized by specific T cell receptors (TCRs). This recognition leads to T cell activation, characterized by cytokine secretion (IFN-γ, TNF-α) and the induction of cytotoxicity (Perforin and Granzymes). (**b**). NK cell activation: HLA-E complexed with activating peptides (VL9 or other pathogen-derived peptides) engages the activating receptor CD94-NKG2C on Natural Killer (NK) cells, delivering a pro-activation signal. The expansion of NKG2C^+^ NK cells is associated with adaptive or memory-like NK responses, particularly in cytomegalovirus (CMV) infections. (**c**). NK cell inhibition: HLA-E loaded with inhibitory peptides (VL9-HLA-E signal peptides or CMV-derived peptides) interacts with the CD94-NKG2A receptor on NK cells. This interaction reinforces inhibitory signaling, thereby suppressing NK cell activity. Created with BioRender.com (https://app.biorender.com/illustrations/6911a51768668d4a46cd14cb, accessed on 19 November 2025).

**Figure 2 cells-14-01983-f002:**
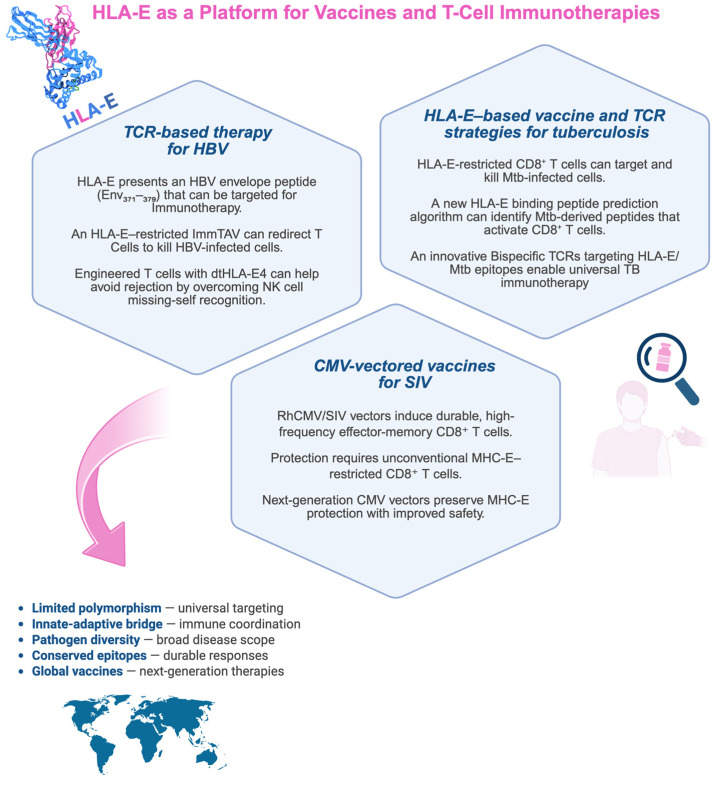
HLA-E–based immunotherapeutic strategies in infectious diseases. The scheme summarizes recent advances in HLA-E-based approaches in immunotherapy and vaccine development for infectious diseases. The figure is divided into three sections, each highlighting a different pathogen (*Mycobacterium tuberculosis*, SIV/HIV, and HBV) and the corresponding immunotherapeutic or vaccine development. HLA-E, a limited polymorphic molecule, presents conserved epitopes derived from various pathogens to CD8^+^ T cells, making it a potential universal target for next-generation therapies. Created with BioRender.com (https://app.biorender.com/illustrations/691efe9c1423118fb89159ce, accessed on 21 November 2025).

**Figure 3 cells-14-01983-f003:**
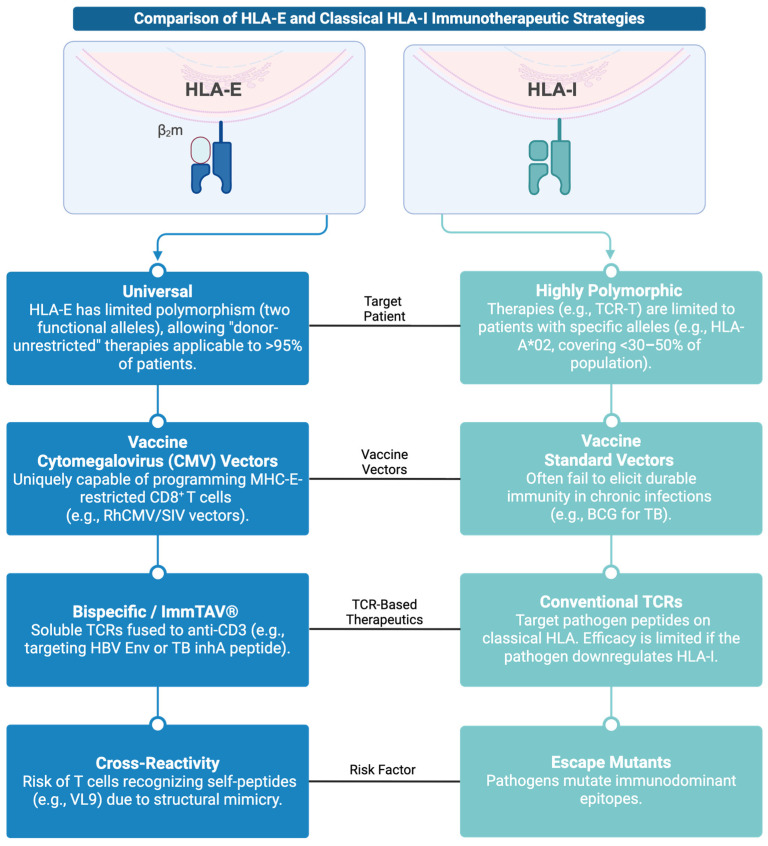
Infographic comparing HLA-E and Classical HLA-I Immunotherapeutic Strategies. Created with BioRender.com (https://app.biorender.com/illustrations/692706e69c2554ccbfa055fe, accessed on 3 December 2025).

**Table 2 cells-14-01983-t002:** A representative list of known HLA-E-restricted epitopes.

Origin	Peptide Sequence	Length	Name	Reference
CMV	VMAPRTVLL	9	VLL	[31,49,61,62,117,118]
CMV	VTAPRTLLL	9	T2-LLL	
CMV	VTAPRTVLL	9	T2-VLL	
CMV	VMAPRTLLL	9	LLL	
CMV	VMAPRTLVL	9	LVL	
CMV	VMAPRTLIL	9	LIL	
SARS-CoV-2	VMPLSAPTL	9	Nsp13	[54]
SARS-CoV-2	VMPLSAPTL	9	NSP13	[23]
SARS-CoV-2	VMYASAVVL	9	NSP6	
SARS-CoV-2	YLQPRTFLL	9	SPIKE	
SARS-CoV-2	MMISAGFSL	9	NSP14	
SARS-CoV-2	YQPYRVVVL	9	SPIKE	
MTB	VLRPGGHFL	9	p68	[35,48,102,109]
MTB	RMPPLGHEL	9	p62	
MTB	VMATRRNVL	9	p55	
MTB	FLLPRGLAI	9	p54	
MTB	RLPAKAPLL	9	p44	
MTB	VMTTVLATL	9	p34	
MTB	EIEVDDDLIQK	11	Rv0634A19-29	
HIV	TALSEGATP	9	TP9	[100]
HIV	RIRTWKSLV	9	RV9	
HIV	RMYSPVSIL	9	V6-RL9	
HIV	PEIVIYDYM	9	PM9	
HIV	RMYSPTSIL	9	RL9	[16]

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
