# Peer review of "The Expanding Role of HLA-E in Host Defense: A Target for Broadly Applicable Vaccines and Immunotherapies"

_cells, 2025, doi:10.3390/cells14241983_

Round 1

Reviewer 1 Report

Comments and Suggestions for Authors

A few points deserve clarification:

line 60-the authors should state early in the paper that HLA-E interacts with both an inhibitory (NKG2A) and an activating (NKG2C) receptor on NK cells. The role of NKG2C is not discussed in detail in the manuscript. 

lines 344-345- the need for transfection of the TCRs is not clear;the CD8+ HLA-E restricted cell lines already recognize HIV-1

line 374-what is the significance of KIRDL1 negativity. KIRs are not discussed at all in this paper

line 471-HLA-E restricted CD4+ T cells? Please clarify

line 610-MHC-II restricted CD8+ T cells? Please clarify

line 648- dtHLA-E, please expand description

line 653-ImmTAV, please expand descriptions

other comment--HLA-E exists in soluble form. what are the implications for any of the infectious diseases reviewed in the paper and for vaccine development

Reviewer 2 Report

Comments and Suggestions for Authors

Title of the paper: The Expanding Role of HLA-E in Host Defense: A Target for Broadly Applicable Vaccines and Immunotherapies

This is an interesting review which provides an overview of the non-classical MHC class Ib molecule HLA-E, emphasizing its dual role in innate and adaptive immunity. It summarizes recent advances in understanding how HLA-E interacts with NK-cell receptors (CD94/NKG2A and CD94/NKG2C) to regulate cytotoxic activity, presents pathogen-derived peptides to CD8⁺ T cells, generating unconventional cytotoxic responses and plays important roles in a variety of viral (CMV, SARS-CoV-2, HBV, HIV) and bacterial (Mycobacterium tuberculosis, Salmonella Typhi) infections. The authors also discuss the therapeutic and vaccine potential of HLA-E-restricted immunity, reviewing CMV-vectored vaccines, donor-unrestricted TCR-based immunotherapies, and peptide prediction algorithms that exploit HLA-E’s limited polymorphism and stability. Some of the key contributions of this article include: (a) Consolidation of scattered findings across viral and bacterial infections involving HLA-E, (b) Emphasis on HLA-E as a translational target for broadly applicable vaccines and immunotherapies, (c) Extensive literature coverage including molecular, structural, and functional insights. Overall, the review presents an informative synthesis of HLA-E biology and its translational implications, suitable for readers in immunology, vaccinology, and molecular medicine.

Regarding the methodology, although it is a review, it should still demonstrate methodological rigor in literature selection, data interpretation and analytical synthesis.  The manuscript appears to rely on comprehensive literature inclusion, including as references an extensive set of primary studies and reviews up to early 2025. However, the criteria for literature selection (e.g., inclusion/exclusion strategy, emphasis on recent or high-impact findings) are not described. The authors should outline the literature search strategy (databases, keywords, timeframe) to enhance transparency and reproducibility. The review demonstrates breadth but lacks depth. It mostly summarizes individual studies sequentially instead of comparing, contrasting, and integrating them.  There is minimal critical analysis of conflicting data, particularly regarding TAP dependence of HLA-E peptide loading in CMV infection, whether HLA-E–restricted responses are protective or tolerogenic in M. tuberculosis and the strength of in vitro versus in vivo evidence for HLA-E-based vaccines.  The authors in many times confuse correlation with causation, especially when linking HLA-E upregulation with disease control, without emphasizing that most evidence is correlative.

Regarding the logical flow and organization the pathogen-based structure seems logical, but the sections vary widely in depth and focus. Some paragraphs (e.g., HBV, HIV) are descriptive, while the immunotherapy section lacks integration with earlier findings. Regarding the conclusions, the authors in this section repeat prior points and do not refer to  new conceptual insights or research priorities.

          Mechanistic descriptions of HLA-E interactions with CD94/NKG2A and CD94/NKG2C are accurate. However,  terminology is sometimes inconsistent (e.g., alternating between “non-classical HLA class I” and “HLA class Ib”). Figures are clear but superficial since they illustrate processes rather than relationships or hierarchies.

The authors in the conclusion emphasize HLA-E’s limited polymorphism and translational potential, but they don’t offer insights or synthesis or a future roadmap. There are no specific hypotheses or critical questions posed to guide future work. I believe that this  section would benefit from bullet-pointed research priorities or a model summarizing the continuum from basic biology to clinical application.

Overall major points to be addressed by the authors:

  • Add critical synthesis: Move beyond summarization. Discuss why findings differ, what mechanisms remain unresolved, and how they could be experimentally addressed.
  • Reframe around central questions: Instead of listing pathogens, organize the review around key immunological themes:
  1. Mechanisms of HLA-E peptide presentation
  2. Functional consequences (activation vs. inhibition)
  3. Implications for immune evasion
  4. Translational strategies (vaccines, TCR-based therapies)
  • Clarify translational bridge: Explicitly connect HLA-E biology to clinical strategies — for example, discuss the design principles of HLA-E–based vaccines and potential risks of cross-reactivity.
  • Include a conceptual figure or table:
    • Figure: HLA-E’s position at the interface of innate and adaptive immunity.
    • Table: Summary of known HLA-E–restricted pathogen-derived peptides (pathogen, peptide, immune cell type, function, reference).
  • Revise the conclusion: Add a short “Future Perspectives” section outlining specific priorities such as:
    • Structural definition of HLA-E peptide-binding repertoire.
    • In vivo quantification of HLA-E–restricted T cells in infection and vaccination.
    • Development of peptide prediction algorithms and standardized tetramers.
    • Integration with checkpoint inhibitor or TCR-based therapies.

Minor points also to be addressed:

  • Shorten paragraphs - many exceed 12–15 lines and include excessive citation chains.
  • Balance section lengths - bacterial infections deserve equal analytical depth as viral ones.
  • Standardize terminology - use either “HLA class Ib” or “non-classical HLA class I” consistently.
  • Improve flow with transition sentences linking innate and adaptive immune roles.
  • Edit figures to emphasize conceptual relationships, not merely mechanisms.
  • Use more active voice to improve readability.
  • Avoid redundant phrasing (“it is essential to understand and clarify…” → “it is important to understand…”).
  • Review grammar, punctuation, and capitalization (especially in technical abbreviations).
  • Verify BioRender license statements and citation formatting per MDPI requirements.
  • Clarify conflict-of-interest statement, given that one author is an editor of Cells.

Round 2

Reviewer 2 Report

Comments and Suggestions for Authors

The manuscript has been substantially improved since the last version. A few more remarks to be addressed by the authors:

  1. Abstract: The abstract  summarizes the topic but is  dense. Consider shortening the section that lists pathogens and highlighting key advances in immunotherapy. The sentence starting with “This review comprehensively surveys…” could be split for better readability
  2. Introduction: The introduction could be improved by  explaining why HLA-E is particularly important in vaccine design, especially compared to HLA-G or HLA-F. The methodology for literature search is  outlined but it seems more like a methods section; consider shortening or moving to a separate “Literature Search Methodology” box.
  3. Characteristics of HLA-E:  The contrast between TAP-dependent and independent loading could be condensed to reduce duplication. Consider adding a table summarizing key differences between HLA-E*01:01 and *01:03 alleles (expression levels, stability, disease associations).
  4. Section 3: Activation and Inhibition: Figure 1 is complex; explanation in the text does not adequately guide readers through its components. Add more discussion on quantitative expression levels of NKG2A vs. NKG2C on various cell types (NK, γδ T, CD8), as briefly mentioned.
  5. Section 4: Viral Infections:
    1.  CMV: The discussion on soluble HLA-E as a biomarker is insightful but deserves deeper analysis on potential clinical validation. Suggest clarifying whether sHLA-E reflects general inflammation or virus-specific mechanisms.
    2. Sars-CoV-2: There is some repetition regarding NK cell exhaustion and NKG2A-mediated inhibition; consider consolidating these points.
    3. HBV:  The section is dense and shifts quickly between NK and T cell mechanisms -recommend clearer separation of themes.
    4. HIV: Some contradictory findings are presented (e.g., on whether HIV peptides reliably bind HLA-E), but there is little critical analysis to explain these discrepancies. Recommendation: include a comparative summary of strong vs. weak HLA-E-binding HIV peptides.
  6. Section 5: Bacterial Infections
    1. M. tuberculosis: Consider adding a diagram or table summarizing identified HLA-E/Mtb peptides and their phenotypic characteristics (Th1 vs regulatory profiles).
    2. Salmonella typhii: Could elaborate more on functional readouts (e.g., perforin, granzyme, memory subsets).
  7. Table 1: Very useful, but the column “Immune cell type/Function” is inconsistent in detail. Some entries discuss NK inhibition, others T-cell activation, and some are vague. Standardize for clarity. Authors could group peptides by disease (CMV, SARS-CoV-2, HIV, Mtb) to enhance usability.
  8. Section 6: Immunotherapy & Vaccines:  Authors could add a small table comparing HLA-E vs classical HLA-I-based immunotherapies.
  9. Figure 1: Dense and text-heavy; consider abbreviating labels and increasing contrast between activating and inhibitory pathways.
  10. Conclusion and Future Perspectives: Well-written and forward-looking, but a bit repetitive with earlier claims. Consider merging with “Future Perspectives” or reducing redundancy. Could benefit from a schematic highlighting major knowledge gaps (e.g., HLA-E epitope prediction, T-cell exhaustion dynamics).
Comments on the Quality of English Language
  • Minor grammatical issues: e.g., “in the absence or suppression of classical HLA pathways” could be rephrased for clarity.
  • Be consistent with formatting: Some sections use “CMV”, others “HCMV”.
  • Also, superscripts for CD8⁺ and CD4⁺ should be consistent.
